# A population-based cohort study of obesity, ethnicity and COVID-19 mortality in 12.6 million adults in England

Thomas Yates [1,2✉], Annabel Summerfield[3], Cameron Razieh [1,2,4], Amitava Banerjee [5,6], Yogini Chudasama[1,4], Melanie J. Davies[1,2,7], Clare Gillies[1,4,8], Nazrul Islam [9], Claire Lawson[4], Evgeny Mirkes [10], Francesco Zaccardi[1,4], Kamlesh Khunti[1,4,7,8,12] & Vahé Nafilyan[3,11,12]

Obesity and ethnicity are known risk factors for COVID-19 outcomes, but their combination has not been extensively examined. We investigate the association between body mass index (BMI) and COVID-19 mortality across different ethnic groups using linked national Census, electronic health records and mortality data for adults in England from the start of pandemic (January 2020) to December 2020. There were 30,067 (0.27%), 1,208 (0.29%), 1,831 (0.29%), 845 (0.18%) COVID-19 deaths in white, Black, South Asian and other ethnic minority groups, respectively. Here we show that BMI was more strongly associated with COVID-19 mortality in ethnic minority groups, resulting in an ethnic risk of COVID-19 mortality that was dependant on BMI. The estimated risk of COVID-19 mortality at a BMI of 40 kg/m$^2$ in white ethnicities was equivalent to the risk observed at a BMI of 30.1 kg/m$^2$, 27.0 kg/m$^2$, and 32.2 kg/m$^2$ in Black, South Asian and other ethnic minority groups, respectively.

---

[1] Diabetes Research Centre, University of Leicester, Leicester General Hospital, Leicester LE5 4PW, UK. [2] National Institute for Health Research (NIHR) Leicester Biomedical Research Centre (BRC), Leicester General Hospital, Leicester LE5 4PW, UK. [3] Office for National Statistics, Newport, UK. [4] Leicester Real World Evidence Unit, Diabetes Research Centre, University of Leicester, Leicester, UK. [5] Institute of Health Informatics, University College London, London, UK. [6] Department of Cardiology, Barts Health NHS Trust, London, UK. [7] Leicester Diabetes Centre, University Hospitals of Leicester NHS Trust, Leicester General Hospital, Leicester, UK. [8] NIHR Applied Research Collaboration – East Midlands (ARC-EM), Leicester General Hospital, Leicester, UK. [9] Nuffield Department of Population Health, Big Data Institute, University of Oxford, Oxford, UK. [10] Department of Mathematics, University of Leicester, Leicester, UK. [11] Faculty of Public Health, Environment and Society, London School of Hygiene and Tropical Medicine, London, UK. [12] These authors contributed equally: Kamlesh Khunti, Vahé Nafilyan. ✉email: ty20@leicester.ac.uk

O besity has emerged as one of the most characterised risk factors internationally for coronavirus disease 2019 (COVID-19) severity and mortality in both community and in-patient settings[1–7]. The strong association between obesity and COVID-19 outcomes has been suggested to result from a deleterious change in the role of circulating adipocytokines leading to a pro-inflammatory state with subsequent predisposition to thrombosis, incoordination of innate and adaptive immune responses, inadequate antibody responses, and the cytokine storm[1].

There is growing evidence that the strength of association between BMI and COVID-19 outcomes may be modified by key sociodemographic factors, most notably ethnicity[6,7], which is also an important risk factor of COVID-19 severity and mortality, with risk up to four times greater in Black and South Asian ethnicities[8–10]. In a study of 65,932 in-patients admitted with COVID-19[7], a coding of obesity was associated with a higher risk of intensive care, mechanical ventilation or in-hospital mortality in all ethnic groups, but with the greatest risk observed in Black ethnicities with obesity[7]. A community study of 6.9 million adults from general practices in England also found the association between BMI and COVID-19 mortality at the start of pandemic was strongest in Black ethnicities[6]. However, whilst ethnicity has been shown to modify associations between BMI and COVID-19 outcomes, previous research has not quantified how this interaction affects both within-ethnicity and between-ethnicity risk across the spectrum of BMI. An early analysis of 5,623 community and in-hospital test results suggested the potential importance of this by showing the risk of SARS-CoV-2 positivity was not different between ethnic groups at low BMI, but was over twofold higher in ethnic minority groups compared to white ethnicities at high BMI[11]. This has not been explored in larger representative community cohorts or with COVID-19 outcomes.

Previous analyses with cardiometabolic outcomes have used the differential associations between ethnicity and BMI to calculate thresholds for obesity in ethnic minority groups where risk is equivalent to white ethnicities at established thresholds for obesity (e.g. 30 kg/m$^2$)[12–14], with current guidelines suggesting that thresholds for ethnic minority groups should be reduced by 2.5 kg/m$^2$[15,16]. It is unclear whether these guidelines are applicable to COVID-19 outcomes. Therefore elucidating the within and between ethnicity risk with COVID-19 mortality has important implications for public health policy and guidelines in relation to infectious disease.

The aim of this study was to use linked national Census, electronic health care records and mortality datasets to investigate the interaction between BMI and ethnicity in the risk of COVID-19 mortality, quantify how the difference in risk between ethnic groups varies by BMI, and generate risk equivalency at established BMI thresholds for class I, II, and III obesity.

## Results

**Cohort characteristics.** This analysis included 11,074,708 (53.6% women, 61.9 [±13.4] years) white, 416,542 (57.3% women, 56.4 [±11.7] years) Black, 621,691 (51.0% women, 55.7 [±12.4] years) South Asian and 478,196 (54.9% women, 55.3 [±11.6] years) from other ethnic minority groups with linked BMI data from family practice in England. The full descriptive profile of the cohort stratified by ethnicity is displayed in Supplementary Table 1; characteristics further stratified by ethnicity and BMI category are presented in Supplementary Data 1 (continuous factors) and 2 (categorical factors). Those with and without BMI data are displayed in Supplementary Table 2. Definitions of characteristics used can be found in Table 1. In total there were 33,951 COVID-19 deaths within the population, of which 31,899 (94.0%) were

coded as U07.1. There were 30,067 (0.27%), 1,208 (0.29%), 1,831 (0.29%), 845 (0.18%) COVID-19 deaths in white, Black, South Asian and other ethnic minority groups, respectively.

**Associations of BMI, ethnicity and COVID-19 mortality.** BMI was associated with COVID-19 mortality in all ethnic groups. However, compared to white ethnicities, the J-shaped association were steeper in Black, South Asian and other ethnic minority groups ($P < 0.001$ for interaction) (Fig. 1A, with specific values highlighted in Table 2), such that at a BMI of 40 kg/m$^2$, the hazard ratio (HR) for white, Black, South Asian and other ethnic minority groups were 1.73 (1.59, 1.91), 3.01 (2.32, 3.90), 5.25 (4.06, 6.79) and 3.89 (2.72, 5.54), respectively, compared to the reference of a BMI of 22.5 kg/m$^2$ in white ethnicities (Table 2). The interaction between BMI and ethnicity revealed that differences in the risk of COVID-19 mortality in Black, South Asian and other ethnic minority groups compared to white ethnicities varied substantially with BMI (Fig. 1B, with specific values highlighted in Table 2). At a low BMI of 20 kg/m$^2$, there was no difference in the risk of COVID-19 mortality in Black (HR = 0.95; 0.78, 1.16) or other minority (HR = 1.13; 0.95, 1.34) ethnicities relative to white ethnicities and only a marginally elevated risk in South Asian ethnicities (HR = 1.21; 1.04, 1.41). Whereas at a BMI of 40 kg/m$^2$, the risk in Black, South Asian and other ethnic minority groups relative to white ethnicities had widened to 1.74 (1.35, 2.26), 3.05 (2.36, 3.94), 2.25 (1.58, 3.21) respectively (Table 2).

The equivalent risk of COVID-19 mortality compared to white ethnicities at a BMI of 35 kg/m$^2$ (HR = 1.24; 1.34, 1.14 compared to reference) was observed at BMI values of 25.2 (21.5, 27.6) kg/m$^2$ and 28.7 (26.0, 30.3) kg/m$^2$ in Black and other ethnic minority groups, respectively (Supplementary Fig. 1); risk equivalence was not possible for South Asian ethnicities where COVID-19 mortality risk was elevated even at low BMI. The equivalent risk of COVID-19 mortality in white ethnicities at a BMI of 40 kg/m$^2$ (HR = 1.73; 1.59, 1.91) was observed at a BMI of 30.1 (28.6, 31.9) kg/m$^2$, 27.0 (24.9, 29.4) kg/m$^2$, and 32.2 (30.6, 33.9) kg/m$^2$ in Black, South Asian and other ethnic minority groups, respectively (Supplementary Figure 1). All ethnic minority groups at any BMI value had a higher risk of COVID-19 mortality than white ethnicities at a BMI of 30 kg/m$^2$.

**Associations of BMI, ethnicity and COVID-19 hospitalisation.** There were 84,282 (0.76%), 4696 (1.13%), 7025 (1.13%), 4000 (0.84%) hospital admissions in white, Black, South Asian and other ethnic minority groups, respectively, with associations of ethnicity and BMI with hospital admissions consistent with those observed for COVID-19 mortality (Fig. 2, Table 2).

**Sensitivity and stratified analysis for COVID-19 mortality.** Associations with COVID-19 mortality were similar after further adjustment for clinical factors (model 2 shown in Supplementary Fig. 2). With this further adjustment, an equivalent risk to white ethnicities at a BMI of 40 kg/m$^2$ was observed at BMI values of 29.9 (28.0, 32.0) kg/m$^2$, 26.7 (24.9, 31.1) kg/m$^2$, and 31.7 (30.2, 33.5) kg/m$^2$ in Black, South Asian and other minority ethnicities references, respectively.

Associations between BMI and COVID-19 mortality were consistent across men and women (Supplementary Figs. 3 and 4), but there was evidence that the association was stronger and differences between ethnic groups more pronounced in those under 70 years of age. In this age group, the risk of COVID-19 mortality at a BMI of 40 kg/m$^2$ increased to 2.71 (2.17, 3.38), 6.99 (4.78, 10.22), 9.48 (6.53, 13.75) and 7.95 (5.05, 12.51) in white, Black, South Asian and other ethnic minority groups, respectively

**Table 1 Covariate and model details.**

| Variable | Coding | Model where covariate included |
|---|---|---|
| **Geographical variables** | | |
| Region | Dummy variables representing region of residence within England (South East, London, North West, East of England, West Midlands, South West, Yorkshire and the Humber, East Midlands, North East) | 1,2 |
| Population density of Lower Super Output Area (see table footnote) | Second-order polynomial, allowing for a different slope beyond the 99th percentile of the distribution to account for extreme values | 1,2 |
| Rural urban classification | Rural hamlets and isolated dwellings, Rural hamlets and isolated dwellings in a sparse setting, Rural town and fringe, Rural town and fringe in a sparse setting, Rural village, Rural village in a sparse setting, Urban city and town, Urban city and town in a sparse setting, Urban major conurbation, Urban minor conurbation | 1,2 |
| **Demographic and socio-economic variables** | | |
| Age | Age at December 31st 2019, included as a continuous variable | 1,2 |
| Sex | Woman or man | 1,2 |
| Index of Multiple Deprivation (IMD) | Dummy variables representing deciles of deprivation – from 1 (most deprived) to 10 (least deprived) | 1,2 |
| Household deprivation (see table footnote) | Not deprived, deprived in one dimension, deprived in two dimensions, deprived in three dimensions, deprived in four dimensions | 1,2 |
| Household tenure | Own outright, own with mortgage, social rented, private rented, other | 1,2 |
| Social Grade of the household reference person (see table footnote) | AB Higher and intermediate managerial/administrative/professional; C1 Supervisory, clerical, junior managerial/administrative/professional; C2 Skilled manual workers; D Semi-skilled and unskilled manual workers; E On state benefit, unemployed, lowest grade workers (Based on household tenure for people aged 75 or over) | 1,2 |
| Level of highest qualification | Degree, A-level or equivalent, GCSE or equivalent, no qualification | 1,2 |
| **Household variables** | | |
| Household size | 1–2 people, 3–4 people, 5–6 people, 7+ people | 1,2 |
| Multigenerational household | Dummy for households with at least one person 65+ and someone at least 20 years younger | 1,2 |
| Household with children | At least one child aged 9 to 18 | 1,2 |
| **Occupational exposure variables (see table note)** | | |
| Key worker type | Education & childcare, food & necessity goods, health & social care, public services, national & local government, public safety & national security, transport, utilities & communication, not a key worker | 1,2 |
| Key worker in the household | Yes, no | 1,2 |
| Exposure to disease (see table footnote) | Score ranging from 0 (no exposure) to 100 (maximum exposure), derived from O*NET data | 1,2 |
| Proximity to others (see table footnote) | Score ranging from 0 (no exposure) to 100 (maximum exposure), derived from O*NET data | 1,2 |
| Household exposure to disease | Maximum 'exposure to disease' score within each household | 1,2 |
| Household proximity to others | Maximum of 'proximity to others' score within each household | 1,2 |
| **Health-related variables** | | |
| Chronic kidney disease (CKD) | No CKD, CKD3, CKD4, CKD5 | 2 |
| Learning disability | No learning disability, Down's Syndrome, other learning disability. | 2 |
| Cancer and immunosuppression | Dummies for blood cancer, solid organ transplant, prescribed immunosuppressant medication, prescribed leukotriene or long-acting beta blockers, prescribed regular prednisolone. | 2 |
| Other conditions | Diabetes, Chronic obstructive pulmonary disease (COPD), Asthma, Rare pulmonary diseases, Pulmonary hypertension or pulmonary fibrosis, Coronary heart disease, Stroke, Atrial Fibrillation, Congestive cardiac failure, Venous thromboembolism, Peripheral vascular disease, Congenital heart disease, Dementia, Parkinson's disease, Epilepsy, Rare neurological conditions, Cerebral palsy, Severe mental illness (bipolar disorder, schizophrenia, severe depression), Osteoporotic fracture, Rheumatoid arthritis or Systemic lupus erythematosus, Cirrhosis of the liver. | 2 |

There are 32,844 Lower Super Output Area (LSOA) areas in England, with a mean population of 1500 and a minimum of 1000. We calculated density as LSOA population divided by LSOA area. Household deprivation is defined across four dimensions: employment (at least one household member is unemployed or with long-term sickness, not including full-time students); education (no household member has at least Level 2 education, and no one aged 16–18 years is a full-time student); health and disability (at least one household member reported their health status as being 'bad'/ 'very bad' or has a long-term health problem); and housing (the household's accommodation is overcrowded, with an occupancy rating −1 or less, or is in a shared dwelling, or has no central heating). Approximate Social Grade is a socio-economic classification based on the occupation, employment, qualification, and tenure of the household reference person. Key worker type is defined based on the occupation and industry code. 'Exposure to disease' and 'proximity to others' are derived from the O*NET database, which collects a range of information about individual working conditions linked to specific occupational codes. To calculate the proximity and exposure measures, the questions asked were: (i) How physically close to other people are you when you perform your current job? (ii) How often does your current job require that you be exposed to diseases or infection? Scores ranging from 0 (no exposure) to 100 (maximum exposure) were calculated based on these questions. Health data were extracted from primary care records, apart from solid organ transplant and stroke which were extracted from hospital records.

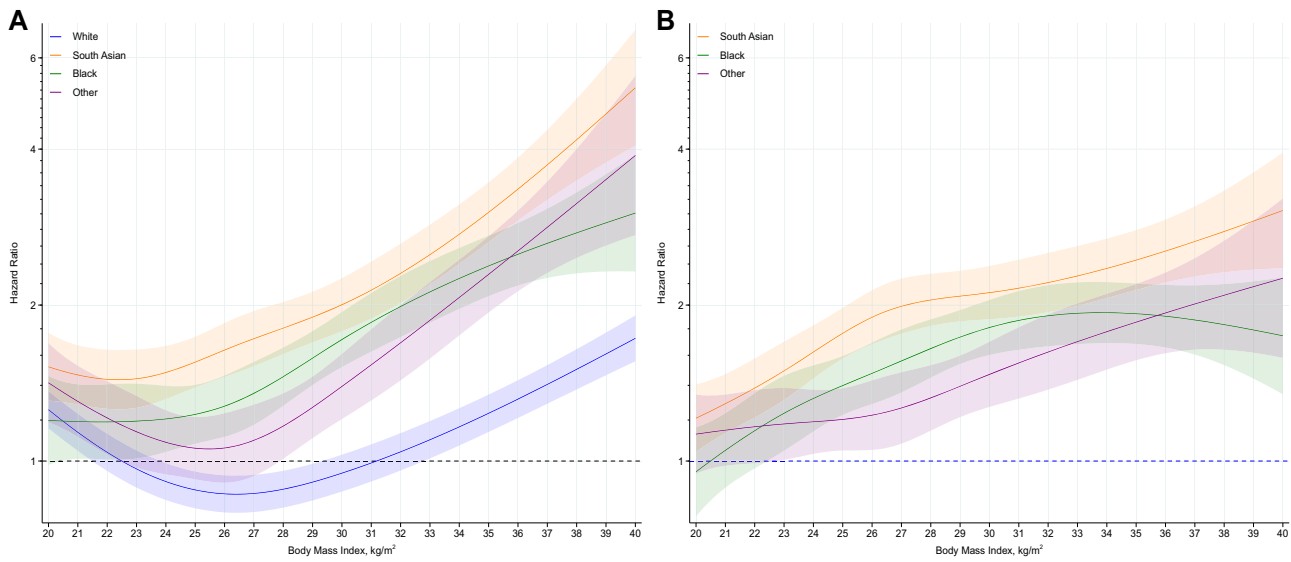

**Fig. 1 Association of BMI and ethnicity with COVID-19 mortality. A** (left): Association of body mass index with COVID-19 mortality stratified by ethnic group, with the reference (HR = 1) placed at body mass index of 22.5 kg/m² for white ethnicities. **B** (right): Hazard of COVID-19 mortality in South Asian, Black and other ethnic minority groups, relative to white ethnicities (HR = 1), across body mass index as a continuous variable. Shaded area as 95% CI. Data adjusted for region, population density, urban/rural classification, deprivation (area and household), social grade, qualification, household size and tenure, household composition (multigenerational, with children), key worker status and type, occupational exposure to disease, occupational exposure to others. The c-index for the model was 0.902, indicating a strong fit.

(model 1), compared to the risk at a BMI of 22.5 kg/m² in white ethnicities (Supplementary Figs. 5 and 6).

When analysis was repeated using a broader range of ethnic classification, the pattern of results mirrored the main finding with South Asian ethnicities (Bangladeshi and Pakistani) having the greatest risk at higher BMI (Supplementary Fig. 7). White ethnicities had the lowest risk. However, Chinese ethnicities may not reflect the wider trend for other ethnic minority groups, with associations similar to white ethnicities.

### Discussion

In 12.6 million adults with linked Census, electronic health care records and mortality data, BMI was associated with COVID-19 mortality amongst all ethnic groups, but with a stronger association in ethnic minority groups. The interaction between BMI and ethnicity with COVID-19 mortality revealed an ethnic risk that was also dependant on BMI. There was no difference in risk between Black and other ethnic minority groups compared to white ethnicities at a low BMI of 20 kg/m², and only a modestly elevated risk in South Asians (HR = 1.21; 1.04, 1.41). However, the risk of COVID-19 mortality in Black, South Asian and other ethnic minority groups, compared to white ethnicities, became more pronounced at higher BMI, with ethnic minority groups reaching between 1.74 (1.35, 2.26) (Black) to 3.05 (2.36, 3.94) (South Asian) times greater risk compared to white ethnicities at a BMI of 40 kg/m². A similar pattern of association was observed for the risk of hospital admissions. The pattern of association between BMI and COVID-19 mortality across ethnic groups produced an equivalent level of risk at substantially different BMI values; for example, the risk of COVID-19 mortality observed in white ethnicities at a BMI of 40 kg/m² was equivalent to the risk observed at BMI values of 30.1, 27.0 and 32.2 kg/m² in Black, South Asian and other ethnic minority groups.

This is the first large-scale population-based study to show the continuous association between BMI and COVID-19 mortality across different ethnic groups on a population level, and to

provide BMI values that show equivalent risk at commonly used thresholds for obesity classifications. Our findings are consistent with previous observations in a community setting at the start of the pandemic and a later in-hospital study[6,7], which also observed an interaction between BMI and ethnicity with COVID-19 outcomes. We extend these previous studies by quantifying the shape of the interaction across a continuous measure of BMI using linked Census and health care records up to the end of 2020, which allowed for the adjustment of detailed sociodemographic characteristics and comorbidities in a population-level dataset. Our findings suggest that, unlike other health outcomes such as type 2 diabetes[12–14], it may not be possible to achieve BMI threshold equivalency in the risk of COVID-19 mortality for class I obesity. Current guidelines suggest that BMI thresholds for obesity classifications should be reduced by 2.5 kg/m² in ethnic minority groups[15,16]. This study suggests that applying these criteria to COVID-19 mortality will only have a marginal impact and still produce thresholds where risk is substantially elevated in ethnic minority groups compared to white ethnicities.

The shape of association between BMI and COVID-19 mortality or hospital admissions was J shaped, particularly in white and other ethnic minority groups, suggesting that the positive association between BMI and COVID-19 outcomes do not extend to lower levels of BMI where low BMI may also be associated with an elevated risk. This is consistent with meta-analyses for all-cause mortality which have reported the nadir in risk occurs between a BMI of 25 to 30 kg/m²[17,18]. The shape of association in the present study could be explained by the fact that low levels of BMI are associated with malnutrition and higher levels of frailty and sarcopenia[19], which are in themselves associated with a greater risk of COVID-19[20,21]. The finding that the association between BMI and COVID-19 mortality was stronger in those under 70 years of age is consistent with previous observations from the United States and Europe and provides further evidence for the importance of BMI as a risk factor in younger populations[5–7]. It is plausible that weaker associations between

**Table 2 Modelled within and between ethnic risk of COVID-19 mortality and hospital admission at specific BMI values.**

**COVID-19 mortality**

| BMI value (kg/m²) | Mortality risk within each ethnicity compared to a single reference category[a] | | | | Mortality risk for Black, South Asian and other ethnic minority groups, compared to white ethnicities[b] | | | |
|---|---|---|---|---|---|---|---|---|
| | White | Black | South Asian | Other ethnic minority groups | White | Black | South Asian | Other ethnic minority groups |
| 20 | 1.26 (1.37, 1.56) | 1.20 (0.98, 1.46) | 1.52 (1.31, 1.77) | 1.42 (1.19, 1.69) | 1 (reference) | 0.95 (0.78, 1.16) | 1.21 (1.04, 1.41) | 1.13 (0.95, 1.34) |
| 22.5 | 1 (reference) | 1 (reference) | 1.44 (1.26, 1.64) | 1.17 (1.00, 1.38) | 1 (reference) | 1.19 (1.01, 1.41) | 1.44 (1.26, 1.64) | 1.17 (1.00, 1.38) |
| 25 | 0.88 (0.81, 0.95) | 1.23 (1.08, 1.40) | 1.55 (1.39, 1.74) | 1.06 (0.92, 1.22) | 1 (reference) | 1.40 (1.23, 1.59) | 1.76 (1.58, 1.98) | 1.20 (1.05, 1.38) |
| 30 | 0.95 (0.87, 1.03) | 1.72 (1.52, 1.94) | 2.00 (1.78, 2.25) | 1.39 (1.21, 1.61) | 1 (reference) | 1.81 (1.60, 2.05) | 2.11 (1.88, 2.38) | 1.47 (1.27, 1.70) |
| 35 | 1.24 (1.14, 1.34) | 2.38 (2.08, 2.72) | 3.02 (2.64, 3.45) | 2.29 (1.94, 2.72) | 1 (reference) | 1.92 (1.68, 2.20) | 2.44 (2.14, 2.79) | 1.85 (1.57, 2.20) |
| 40 | 1.73 (1.56, 1.91) | 3.01 (2.32, 3.90) | 5.25 (4.06, 6.79) | 3.89 (2.73, 5.54) | 1 (reference) | 1.74 (1.35, 2.26) | 3.05 (2.36, 3.94) | 2.25 (1.58, 3.21) |

**Hospital admissions for COVID-19**

| BMI value (kg/m²) | Risk of hospitalisation within each ethnicity compared to a single reference category[a] | | | | Risk of hospitalisation for Black, South Asian and other ethnic minority groups, compared to white ethnicities[b] | | | |
|---|---|---|---|---|---|---|---|---|
| | White | Black | South Asian | Other ethnic minority groups | White | Black | South Asian | Other ethnic minority groups |
| 20 | 1.16 (1.11, 1.21) | 1.19 (1.05, 1.35) | 1.36 (1.24, 1.49) | 1.12 (1.00, 1.24) | 1 (reference) | 1.02 (0.90, 1.16) | 1.17 (1.07, 1.28) | 0.96 (0.86, 1.07) |
| 22.5 | 1 (reference) | 1.31 (1.19, 1.43) | 1.43 (1.34, 1.54) | 1.23 (1.13, 1.34) | 1 (reference) | 1.31 (1.19, 1.43) | 1.43 (1.34, 1.54) | 1.23 (1.13, 1.34) |
| 25 | 0.97 (0.93, 1.01) | 1.38 (1.29, 1.48) | 1.64 (1.55, 1.74) | 1.39 (1.30, 1.49) | 1 (reference) | 1.43 (1.33, 1.53) | 1.70 (1.60, 1.80) | 1.44 (1.35, 1.54) |
| 30 | 1.18 (1.13, 1.23) | 1.93 (1.81, 2.04) | 2.50 (2.36, 2.65) | 2.15 (2.02, 2.29) | 1 (reference) | 1.63 (1.54, 1.74) | 2.12 (2.01, 2.25) | 1.83 (1.71, 1.95) |
| 35 | 1.56 (1.50, 1.62) | 2.72 (2.91, 2.54) | 3.56 (3.34, 3.80) | 2.92 (2.73, 3.18) | 1 (reference) | 1.74 (1.63, 1.86) | 2.29 (2.15, 2.44) | 1.89 (1.75, 2.04) |
| 40 | 2.08 (1.98, 2.19) | 3.42 (3.04, 3.85) | 4.66 (4.11, 5.28) | 3.38 (2.85, 4.00) | 1 (reference) | 1.64 (1.46, 1.85) | 2.24 (1.98, 2.54) | 1.62 (1.37, 1.92) |

Mortality data derived from Fig. 1. Hospital admission data derived from Fig. 2. Data as hazard ratio (95% CI).
[a]Reports the risk of an event relative to white ethnicities at a BMI of 22.5 kg/m². Values therefore represent the modelled risk at specific BMI values in different ethnic groups relative to this single reference point.
[b]Reports the risk of an event relative to white ethnicities at specific BMI values. Therefore at each BMI value the modelled risk is provided for Black, South Asian and other ethnic minority groups relative to white ethnicities.
All data adjusted for region, population density, urban/rural classification, deprivation (area and household), social grade, qualification, household size and tenure, household composition (multigenerational, with children), key worker status and type, occupational exposure to disease, occupational exposure to others.

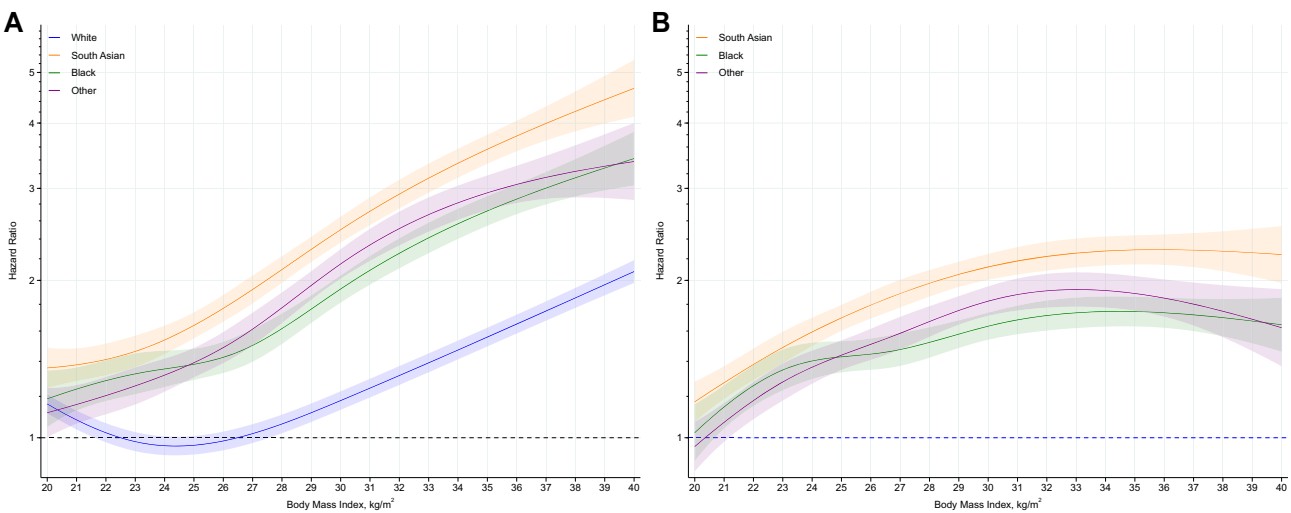

**Fig. 2 Association of BMI and ethnicity with COVID-19 hospital admission. A** (left): Association of body mass index with COVID-19 hospital admission stratified by ethnic group, with the reference (HR = 1) placed at body mass index of 22.5 kg/m² for white ethnicities. **B** (right): Hazard of COVID-19 hospital admission in South Asian, Black and other ethnic minority groups, relative to white ethnicities (HR = 1), across body mass index as a continuous variable. Shaded area as 95% CI. Data adjusted for region, population density, urban/rural classification, deprivation (area and household), social grade, qualification, household size and tenure, household composition (multigenerational, with children), key worker status and type, occupational exposure to disease, occupational exposure to others. The c-index for the model was 0.787 indicating a good fit.

BMI and COVID-19 in older individuals may reflect the greater absolute risk of COVID-19 with age and the risk profile in older normal or underweight weight individuals with frailty or other factors[5,22].

The reasons underpinning the observed ethnicity by obesity interaction are unclear. It has previously been suggested that ethnic minority groups may have a stronger innate inflammatory response to viral infection or chronic disease[23–25], thus potentially increasing the risk of severe COVID-19[23]. It is possible that the presence of greater levels of adiposity interacts with and accelerates this inflammatory response in ethnic minority groups[7]. However, unlike previous findings from the start of the pandemic or from hospital settings where the risk with obesity was found to be greatest in Black ethnicities[6,7], this research using national level data from primary care during the first year of the pandemic suggests that the risk with obesity is greater in all minority ethnic groups compared to white ethnicities, with South Asian ethnicities at greatest risk. This mirrors the interaction between ethnicity and BMI with cardiometabolic disease where risk has also consistently been shown to be greatest in South Asian ethnicities[12–14]. Further research in this area, including the potential of genetic and epigenetic factors, is warranted.

A major strength is the large population-level dataset linking national Census and health care record data making it the largest analysis of its kind to date. Linkage between clinical records and Census data allowed for the extraction of BMI from primary care records and ethnicity from Census data, which is a major strength as ethnicity is not universally coded within primary care[26], with a previous study investigating COVID-19 risk factors in England finding ethnicity coding was missing in over 25% of clinical records[10]. We were also able to extract detailed descriptive and covariate data from a wide range of sociodemographic and clinical factors, allowing for the adjustment of potentially confounding variables including household and area indicators of deprivation and established clinical risk factors for COVID-19 mortality. However, there were limitations. Most notably, this analysis is generalisable to the 52.4% of the English population with coded BMI data within their health care records in the 10

years preceding the pandemic. In England, height and weight are collected as part of routine care. Nevertheless, family practice incentivisation schemes and differential take-up rates to population-level vascular screening programmes means that data are not missing at random[27]. Previous analysis has shown that women, those who attend their family doctor more often, who come from more deprived areas, who have a high or low BMI and have a greater number of comorbidities are more likely to have a coded BMI value[27]. Nevertheless, the pattern of overweight and obesity in this study (66.9% for white ethnicities, 77.4% for Black ethnicities, 65.4% for South Asian ethnicities and 59.4% for other ethnic minority groups) were similar to national survey data were the highest rates (consistently above 70%) are reported in Black ethnicities[28]. It has also been demonstrated that complete case analysis excluding missing data within clinical records can provide unbiased estimates of adjusted exposure-outcome associations under a wide range of missing data assumptions[29], particularly when missingness is independent of the outcome, as was demonstrated for this study. In addition, primary care data in England provide some of the most detailed electronic health care records internationally and are routinely used to identify individuals at risk of chronic and infectious diseases, including COVID-19 mortality[30,31], giving this study real-world utility. However, as this is real word administrative data, it is possible not all COVID-19 deaths or hospital admissions were captured, or conversely, some deaths or hospital admissions may have been coded as attributable to COVID-19 in error. It is notable though that a high proportion of COVID-19 deaths (94.0%) were coded as U07.1, therefore relatively few deaths were subject to symptom-based or epidemiological diagnosed cases. Although we report a secondary outcome of hospital admissions as a marker of disease severity, data was not available for in-hospital treatment. This study utilised data from the 2011 Census, therefore any sociodemographic changes within the last decade will not be reflected in the analysis. Although we adjusted for factors related to the risk of SARS-CoV-2 exposure, including household composition, key worker status and exposure to others, it is not possible to verify whether the associations observed with BMI and

ethnicity were due to greater disease severity, greater SARS-CoV-2 exposure and infection rates, or a combination of both. Therefore results should be interpreted simply as the population-level risk of dying from COVID-19 during the first year of pandemic.

In conclusion, this study of linked Census, electronic health records and mortality data demonstrated a notable interaction between ethnicity and obesity in the risk of COVID-19 mortality and hospitalisation, with obesity having a stronger association in all ethnic minority groups compared to white ethnicities. These results further emphasise the importance of public health messages to reduce levels of obesity within the population, particularly within ethnic minority groups. Future work is needed to investigate how these risk factors interact with post COVID-19 vaccination infection and mortality risk.

## Methods

**Populations and databases**. Ethical approvals for the research were obtained from the University of Leicester, UK (reference 0818UE). This analysis uses data from the Office of National Statistics Public Health Research Database (PHRD), a new linked dataset using the 2011 Census data; all adults within England are required to complete and return the Census data by law, with a response rate of 93.9%[32]. The 2011 Census was linked to the General Practice Extraction Service Data for Pandemic Planning and Research (GDPPR) which contains primary care records for all individuals living in England on November 1st 2019, with records being extracted up to December 31, 2019. This dataset was further linked to mortality records and Hospital Episode Statistics (HES). Linkage between clinical and Census datasets was enabled through NHS numbers that are unique to each individual. To obtain NHS numbers for the 2011 Census, the 2011 Census was linked to the 2011-2013 NHS Patient Registers. It was first linked deterministically using 24 different matching keys. Probabilistic matching (Felligi–Sunter method) was then used to match records that were not linked deterministically, using 13 different combinations of personal identifiers[33].

Our analysis was restricted to those over 40 years of age on December 31, 2019 due to poor coverage of BMI values in GDPPR in younger populations.

Of the 32,755,633 people enumerated at the 2011 Census in England and Wales aged ≥40 years on December 31st 2019, 31,498,128 people were linked deterministically or probabilistically to the NHS Patient register, and of these, 27,477,607 individuals were alive on 24th January 2020. As linked family practice data was only available for England, the English population with linked GDPPR data included 24,026,950 people (see sample flow diagram in Supplementary Fig. 8). Of these, 12,591,137 (52.4%) had valid BMI data and were included within the primary analysis.

**Exposure**. In England, height and weight are collected during routine primary care consultations by trained staff using medical grade equipment with BMI calculated (weight(kg)/height(m)$^2$) and coded as a continuous value within electronic health care records. For this study, BMI was available within the GDPPR extract of primary care records, reflecting the BMI value coded within primary care that was closest to December 31, 2019, with data available from January 2010. Participants without a recorded BMI in primary care within this 10-year window were coded as missing. In order to remove outliers and potentially spurious values, a data-driven approach was used, restricting the analysis from the 2.5th (17.4 kg/m$^2$) to 97.5th (41.0 kg/m$^2$) percentile of the distribution.

**Outcome**. COVID-19 related death (either in hospital or out of hospital) was the primary outcome for the analysis and defined as confirmed or suspected COVID-19 death, which was identified by ICD-10 codes U07.1 (lab-confirmed COVID-19) or U07.2 (clinically/epidemiologically-diagnosed COVID-19 when a lab-confirmed test is inconclusive or not available) anywhere on the death certificate from 24 January 2020 until December 28, 2020.

Hospitals admissions for COVID-19, using a primary admission for COVID-19 (U07.1 or U07.2) were also extracted from HES from 24 January 2020 until December 28, 2020 and were included as an indicator of disease severity as a secondary outcome, as has been reported for other studies[34].

**Effect modifier**. Self-reported ethnicity was coded from the 2011 Census, which asked respondents to select their ethnicity from 18 categories. For the purposes of this analysis we derived four categories: white (defined as British, Irish, other White), South Asian (Asian/Asian British defined as Indian, Pakistani, Bangladeshi), Black (defined as Black African, Black Caribbean, Black British, other Black) or other (all other classifications). Ethnicity was imputed in 3.0% of 2011 Census returns due to item non-response using nearest-neighbour donor imputation, the methodology employed by the Office for National Statistics across all 2011 Census variables[33].

**Covariates**. Our analysis included key Census extracted sociodemographic data, including measures of household deprivation, household tenure and composition, occupation status (including key worker status) and social grade, educational attainment and exposure to disease or others, defined within Table 1. We also used the linkage to GDPPR and HES to extract up-to-date geographical information (including population density and area deprivation) and data on chronic diseases that have been shown to be associated with COVID-19 outcomes in the QCovid® prediction models[30] (Table 1).

**Statistical analysis**. Cox proportional hazard models were fitted with time to event measured in days from 24 January 2020 to the date of COVID-19 deaths or deaths from other causes or December 28, 2020, whichever came first. Non-COVID-19 mortality was analysed as a censoring event. A priori covariates were adjusted for in two models. Model 1 was adjusted for age, sex, ethnicity, geographic region, and other key sociodemographic factors (detailed in Table 1). Model 2 was adjusted for the same factors as Model 1, plus included clinical factors (Table 1). A BMI by ethnicity interaction term was included in both models. Given the potential for included clinical factors to act as mediators between BMI and COVID-19 mortality, the primary interpretation from the analysis was taken from Model 1. The proportional hazards assumption was assessed visually using log-log survival plots across quartiles of BMI. The strength of interaction was tested using a likelihood-ratio test. Restricted cubic splines were fitted with 3 knots at the 25th (23.2 kg/m$^2$), 50th (26.3 kg/m$^2$) and 75th (29.8 kg/m$^2$) BMI percentiles. A BMI of 22.5 kg/m$^2$ (representing a value within the normal range) in white ethnicities as the largest group was specified as the reference to which all other ethnic minority groups and BMI values were compared. Model fit was determined using the concordance statistic (c-index), with values over 0.8 interpreted as a strong model fit. Models were repeated for the outcome of hospital admissions to investigate whether associations of ethnicity and BMI with hospital admissions were consistent with the pattern of associations observed for mortality.

For descriptive purposes, values generated by the restricted cubic spline models were used to quantify the within and between ethnic risk in COVID-19 outcomes at specific BMI values (20, 22.5, 25, 30, 35, 40 kg/m$^2$). For COVID-19 mortality this data was also used to generate BMI values in minority ethnicity groups that would produce an equivalent risk to white ethnicities at the thresholds for class I (30 kg/m$^2$), II (35 kg/m$^2$), and III (40 kg/m$^2$) obesity.

When fitting the Cox models, we included all individuals who died (or were admitted to hospital when using hospital admission as an outcome) during the analysis period and a weighted random sample of those who did not, with a sampling rate of 1% for those of white British ethnicity and 10% for adults from ethnic minority groups. We applied case weights (defined as the inverse of the sampling rate) to all analyses.

In order to assess the pattern of results across sex and age, analyses for COVID-19 mortality were repeated stratified by sex and age (<70 years, ≥70 years). In order to assess whether the pattern of results for the broad ethnic categories of white, Black, South Asian and other mirrored the pattern of results in more detailed sub-categories, the analysis was repeated using ten categories of ethnicity.

As BMI is likely to be missing not at random and influenced by many factors[27], not all of which were captured in this analysis, multiple imputation of missing data was not attempted. Nevertheless, to assess whether the pattern of missingness varied by ethnic groups, we examined the proportion of missing data by ethnicity across regions. There was no clear systematic pattern of missingness by ethnicity (Supplementary Fig. 9). We also undertook logistic regression to quantify whether ethnicity, covariates or outcome predicted missing data using the pseudo R$^2$ or area under the curve statistic (Supplementary Table 3). Missing data was found to be conditionally independent of the outcome indicating a lower risk of bias with the complete case analysis[29].

Data are reported as mean (± SD) or hazard ratio (95% CI) unless detailed otherwise.

**Reporting summary**. Further information on research design is available in the Nature Research Reporting Summary linked to this article.

## Data availability

Analysed data are controlled by the Office of National Statistics, UK. Technical details of the Public Health Research Database (PHRD) incorporating the 2011 Census data for England and Wales, linked to Mortality Data, Hospital Episode Statistics (HES) data, and GP Extraction Service (GPES) data for Pandemic Planning and Research Data can be found through the Health Data Research UK Innovation Gateway https://web.www.healthdatagateway.org/dataset/a325f33e-bac8-49af-896f-1e025941dae8 Given the sensitive nature of the data, organisations and individuals will need to demonstrate they meet strict data security and information governance standards. The application form can be accessed and completed through Health Data Research UK Innovation Gateway https://web.www.healthdatagateway.org/dataset/a325f33e-bac8-49af-896f-1e025941dae8.

## Code availability

The statistical code developed for this study has been archived and published separately [35].

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

## Acknowledgements

This research was funded by a grant from the UKRI (MRC)-DHSC (NIHR) COVID-19 Rapid Response Rolling Call (MR/V020536/1) (T.Y.) and is part of the Data and Connectivity National Core Study, led by Health Data Research UK in partnership with the Office for National Statistics and funded by UK Research and Innovation (HDRUK2020.138) (T.Y., K.K.).

## Author contributions

T.Y., K.K., C.R., V.N. developed the research question. T.Y., C.R., V.N. developed the statistical analysis plan. A.S. undertook the statistical analysis with support from V.N.; both had access to the data. T.Y. drafted the manuscript. T.Y., A.S., C.R., K.K., V.N., A.B., Y.C., M.J.D., C.G., N.I., C.L., E.M., F.Z. contributed to the research design and revised the manuscript for important intellectual content.

## Competing interests

T.Y. is supported by the NIHR Leicester Biomedical Research Centre (BRC) and has received funding from Astra-Zeneca for obesity-related research. M.J.D. is supported by the NIHR Leicester Biomedical Research Centre (BRC) and has received funding from Astra-Zeneca and Novo Nordisk for obesity-related research. K.K. is Director for the University of Leicester Centre for Ethnic Health Research, Trustee of the South Asian Health Foundation, national NIHR Applied Research Collaborations - East Midlands (ARC-EM) lead for Ethnicity and Diversity and a member of SAGE and Chair of the SAGE subgroup on ethnicity and COVID-19. A.B. has research funding separate to this work from Astra-Zeneca and is a Trustee of the South Asian Health Foundation. Other authors declare no conflicts of interest.
