## [Peer Review File · Nature Communications]

A population-based cohort study of obesity, ethnicity and COVID-19 mortality in 12.6 million English adultsREVIEWER COMMENTS

Reviewer #1 (Remarks to the Author):

This is an important manuscript on the association between BMI and COVID-19 death in various ethnic groups in England. This topic was addressed before; however, a clear and major strength of the current study is the large sample size of this cohort.

Comments

Please include to the abstract information on the statistical methods that were used in the analysis and on the confounders that were adjusted for.

Main text

Methods section

Exposure (page 9): Please clarify whether BMI was calculated based on measurements of weight and height or based on self-reports of weight and height.

Outcome (page 9): COVID-19 deaths- please explain the difference between U07.1 and U07.2 codes and discussion potential misclassification of COVID-19 death coding

Covariates (methods and statistical analysis sections pages 9-10)- it is important to provide details within the text of the main manuscript which covariates were analyzed, rather than in supplementary tables. Please present the operational definitions of all covariates in text.

Please provide information on multicollinearity assessment.

Results

Throughout the manuscript, please define whether the numbers next to the sign \pm are standard deviations or standard errors.

COVID-19 mortality might be affected by multiple factors except of BMI. In addition, BMI might be a proxy for other factors; the observed associations might be influenced by confounders. Therefore, it is very important to explicitly state in text the variables that were adjusted for in multivariable models. This is central issue, vague titles such as clinical factors are not sufficient.

Table 1: Please present only one decimal place for means, SD and percentages. Table 1 can be provided as supplementary table.

Given the concerns of potential confounders, it is important to provide a table showing the distribution of confounders according to BMI categories by ethnic group.

Figure 1: please add information on the variables that were included in each model (not only in supplementary material).

Figure 1A and figure 2 show more J shape association rather than dose response association. Figure 1B the same for white and other ethnic groups. Please discussion these observations.

Discussion

Please elaborate on possible explanation regarding the higher COVID-19 mortality at lower BMI among minority groups. What is the role of SES disparities? What is the potential of residual confounders?

The authors state that the use of large census dataset allowed for adjustment for "for detailed sociodemographic characteristics and comorbidities in a population-level dataset". Adjustment for potential confounders is a major concern in observational studies; however the authors did not provide sufficient information in the manuscript on the potential confounders that they adjusted for. Detailed information on these factors should appear in the main text.

Page 5: The authors should elaborate on possible explanations regarding the stronger association between BMI and COVID-19 mortality in people less than 70 years of age.

Reviewer #2 (Remarks to the Author):

Thank you for asking me to review this interesting and potentially important manuscript. The authors use linked UK 2011 census data, primary care data on BMI and co-morbidities to investigate whether the association between BMI and COVID-19 mortality varied with ethnicity. The main findings from data from 12.6M individuals were that BMI was associated with COVID-19 mortality and that there was robust evidence for an ethnicity-BMI interaction, such that the BMI-associated mortality risk was more marked in non-White ethnicities.

The findings are potentially of considerable significance, scientifically and translationally. Although, as the authors note, there have been several studies exploring these RF for severe COVID-19, this is probably the largest and attempts to use population-level data. Overall this is an important paper and the authors should be congratulated on this work. A major advantage is that the data almost completely cover the pre-vaccination period, which is a potential source of bias.

The approach seems well thought out, sound and the analyses appropriate. The weaknesses are largely acknowledged.

I have some comments/queries:

It would be helpful for the mortality rate overall and by ethnic group to be given in the abstract.

Do the HES data include measures of COVID-19 severity, such as need for supplementary oxygen, non-/invasive respiratory support, ICU admission etc? None of these are perfect, but it would obviously be of interest to look at morbidity as well as mortality. This would be of considerable interest and might partly address concerns regarding access/severity at presentation.

BMI data from GPPR were only available on approximately 50% of the cohort and as the authors discuss, this is a potential source of bias. Could the authors give a breakdown of the proportion in BMI categories by ethnicity and how this compares to other data on BMI in UK populations.

There is an impressive and extensive list of covariates included in the models (given in supp data). Were all the models stable with so many covariates and limited deaths in some groups? It would be helpful for some further discussion as to how these were selected for inclusion in the model (many would be correlated). Did the authors consider a model with a limited number of these covariates, including - for example - extant cardiometabolic or respiratory disease that are thought to be RF for COVID-19 severity? A number of the co-morbidities seem less relevant RF. A specific issue in this regard is whether the BMI association is independent of T2D, which is obviously common. Could the authors explore this further?

No definitions are given for the co-morbidities. Are these GP diagnoses and/or hospital diagnoses? Were they verified?

Reviewer #3 (Remarks to the Author):

The paper aims to explore whether the association between BMI and COVID-19 death varies by ethnicity.

A recent paper by Gao et al (Lancet Diabetes and Endocrinology, 2021) uses similar UK electronic health records data to analyse in some detail the links between BMI and COVID outcomes including mortality. The authors of that paper include an analysis of effect modification/interaction by ethnicity, finding evidence of interaction with a steeper BMI-COVID-death association in some non-white groups compared with white. The present paper is therefore not wholly novel. A limitation of the Gao paper is that they only looked at the interaction with the BMI-COVID-19 association specified as linear, whereas the current paper uses non-linear parameterisations and includes a useful visual of the interaction, in Figure 1. Another strength of the present paper is that linked census data were used to provide self-reported ethnicity, which is likely to be more accurate/complete than GP-recorded ethnicity.

I found some of the analysis/presentation of results to be confusing/misleading. The stated aim was to look at how the BMI-COVID-death association varies by ethnicity, but in parts of the results the authors focus on the comparison between ethnic groups at fixed BMI levels, which addresses a different question. In addition, because all hazards are compared against the fixed reference point of "white - 22.5kg/m²", the BMI component is difficult to disentangle from the overall ethnic differences in risk. For example, the authors state that the HR for Black-30kg/m² compared with White-22.5 is 1.72 - but this will include both the effects of BMI and ethnicity, so does not really distill for us how the BMI-COVID-death association is different for Black ethnicity. The whole idea of "BMI of equivalent risk" suffers from the same flaw and I think is misleading. For example, it is stated that the equivalent risk to White-35kg/m² is seen at 25.2kg/m² in Black people. But this is not purely because of a steeper BMI-mortality slope in Black people (as implied by the conclusion that reducing levels of obesity is the solution to achieving equivalent risk) - the entire curve is also shifted substantially upwards due to the ethnicity effect, independently of BMI.

Only 52% of individuals had BMI present - a useful descriptive comparison of those with/without BMI data is provided, confirming missingness to be not completely at random. Missing data is discussed by the authors as a limitation, but it would be good to see more specific discussion of the assumptions being made by the authors' approach of excluding those with missing data (in particular, that missingness is conditionally independent of outcome).

Exclusion of 5% of BMI records (i.e. <2.5th and >97.5th %ile) to avoid outliers seemed extreme to me. What values did these %iles correspond to? Some sensitivity analysis around this would be reassuring.

Death with COVID or suspected COVID anywhere on the death certificate was counted as a COVID-19 death. Was any sensitivity analysis done to explore other definitions (e.g. only confirmed COVID, only as underlying cause of death)?

Figure 2 appears to be essentially a repeat of Figure 1 with some additional annotation; it is not clear that this warrants an extra figure.

Reviewer 1

Reviewer 1 Comment	Authors' Response	Page number
Please include to the abstract information on the statistical methods that were used in the analysis and on the confounders that were adjusted for.	Thank you for this suggestion, we have revised the abstract accordingly adding the statistical methods. Given the number of covariates, for the abstract we have summarised simply as "Sociodemographic factors derived from census records and clinical factors from health care records were included as covariates"	4
Exposure (page 9): Please clarify whether BMI was calculated based on measurements of weight and height or based on self-reports of weight and height.	We have added this to the methods as: "In England, height and weight are collected during routine primary care consultations by trained staff using medical grade equipment with BMI calculated (height (m)/weight(kg) ²) and coded as a continuous value within electronic health care records."	9
Outcome (page 9): COVID-19 deaths- please explain the difference between U07.1 and U07.2 codes and discussion potential misclassification of COVID-19 death coding	U07.1 identifies that lab-confirmed COVID-19 is mentioned as an underlying cause of death U07.2 identifies that clinically/epidemiologically-diagnosed COVID-19 is mentioned as an underlying cause of death when a confirmed test is inconclusive or otherwise not available These descriptions have been added to the text. We have added the below sentence to the Results: "In total there were 33951 COVID-19 deaths within the population, of which 31899 (94.0%) were coded as U07.1." We have also added the below sentence on misclassification to the Discussion "However, as this is routine administrative data, it is possible not all COVID-19 deaths or hospital admissions were captured, or conversely, some deaths or hospital admissions may have been coded as attributable to COVID-19 in error. It is notable though that a high proportion of COVID-19 deaths (94.0%) were coded as U07.1, therefore relatively few deaths were subject to symptom-based or epidemiological diagnosed cases."	9, 13, 19
Covariates (methods and statistical analysis sections pages 9-10)- it is important to provide details within the text of the main manuscript which covariates were analyzed, rather than in supplementary tables.	Given the number and range of covariates included in this analysis, we feel it aids readability to retain this information in a Table format. However, we agree that this information sits better within the main text and have therefore moved the Table to the main text for placement in the Methods section.	10, Table 1

Please present the operational definitions of all covariates in text.		
Please provide information on multicollinearity assessment	Thank you for this informed comment. For the Reviewers information, the variance inflation factor (VIF) for BMI across all spline knots and models was less than 3.5, suggesting only a moderate degree of correlation between BMI and covariates. As a result, only a moderate amount of the variance in BMI can be explained by included covariates.	N/A
Throughout the manuscript, please define whether the numbers next to the sign \pm are standard deviations or standard errors.	We have now added the following to the end of the statistical analysis section to make the use of data clear throughout. "Data are reported as mean (\pm SD) or hazard ratio (95% CI) unless detailed otherwise."	12
COVID-19 mortality might be affected by multiple factors except of BMI. In addition, BMI might be a proxy for other factors; the observed associations might be influenced by confounders. Therefore, it is very important to explicitly state in text the variables that were adjusted for in multivariable models. This is central issue, vague titles such as clinical factors are not sufficient.	We have now moved Table 1 to the main text which displays all covariate information as well as detailing the specific models that each covariate appeared in.	10, Table 1
Table 1: Please present only one decimal place for means, SD and percentages. Table 1 can be provided as supplementary table.	We have changed the number of decimal places and moved to Supplement as suggested.	Supplementary Tables
Given the concerns of potential confounders, it is important to provide a table showing the distribution of confounders according to BMI categories by ethnic group.	Thank you for this suggestion. Given the large amount of data this further stratification of data entails (>2000 table cells), we have uploaded separate excel datasheets with the requested data; we will make this available through an online repository on acceptance for publication.	Uploaded datasheet
Figure 1: please add information on the variables that were	We have added the covariate details to the modified Figure 1	Figure 2

included in each model (not only in supplementary material).		
Figure 1A and figure 2 show more J shape association rather than dose response association. Figure 1B the same for white and other ethnic groups. Please discussion these observations.	We have added the below paragraph to the Discussion: “The shape of association between BMI and COVID-19 mortality or hospital admissions was J shaped, particularly in white and other ethnicities, suggesting that the positive association between BMI and COVID-19 outcomes do not extend to lower levels of BMI where low BMI may even be associated with an elevated risk. This is consistent with meta-analyses for all-cause mortality which have reported the nadir in risk occurs between a BMI of 25 to 30 kg/m² [22,23]. The shape of association in the present study could be explained by the fact that low levels of BMI are associated with malnutrition and higher levels of frailty and sarcopenia [24], which are in themselves associated with a greater risk of COVID-19 [25, 26].”	17
Please elaborate on possible explanation regarding the higher COVID-19 mortality at lower BMI among minority groups. What is the role of SES disparities? What is the potential of residual confounders?	We note the J shaped curve applied to all ethnicities, but was more pronounced in white and other ethnic groups. We hope the above addition to the Discussion also covers this comment. Specifically, it is likely that residual confounding due to malnutrition, frailty and sarcopenia played an important role in these findings across all ethnic groups.	17
The authors state that the use of large census dataset allowed for adjustment for "for detailed sociodemographic characteristics and comorbidities in a population-level dataset". Adjustment for potential confounders is a major concern in observational studies; however the authors did not provide sufficient information in the manuscript on the potential confounders that they adjusted for. Detailed information on these factors should appear in the main text.	Thank you. As we have highlighted in our responses to comments on the covariates above, we have now moved Table 1 to the main text to appear in the Methods section.	10, Table 1

Page 5: The authors should elaborate on possible explanations regarding the stronger association between BMI and COVID-19 mortality in people less than 70 years of age.	We have added the below sentence to the Discussion where the results by age are highlighted: “It is plausible that weaker associations between BMI and COVID-19 in older individuals may reflect the greater absolute risk of COVID-19 with age and the risk profile in older normal or underweight weight individuals with frailty or other factors [5, 27]”	17
---	---	-----------

Reviewer 2

Reviewer 2 Comment	Authors' Response	Page
It would be helpful for the mortality rate overall and by ethnic group to be given in the abstract	Thank you for this suggestion. We have added the below sentence to the abstract: “There were 30,067 (0.27%), 1,208 (0.29%), 1,831 (0.29%), 845 (0.18%) COVID-19 deaths in white, black, South Asian and other ethnic groups, respectively.”	4
Do the HES data include measures of COVID-19 severity, such as need for supplementary oxygen, non-/invasive respiratory support, ICU admission etc? None of these are perfect, but it would obviously be of interest to look at morbidity as well as mortality. This would be of considerable interest and might partly address concerns regarding access/severity at presentation.	We thank the Reviewer for this comment; we have worked to address it in as robust a way as possible with the data we have access to. The Office for National Statistics (ONS) Public Health Data Asset (PHDA) which was used for this study only has agreements in place to access full HES records up to December 31st 2019. Therefore we do not have access to detailed hospital data during the pandemic. However, the PHDA does have agreements in place for linkage to COVID-19 hospital admissions (yes/no) only (i.e. whether or not someone was admitted to hospital for the primary reason of COVID-19). We have added this data to the manuscript as a broad indicator of disease severity as has been reported in other studies (e.g. reference 19 in the text). However, we recognise this has limitations, which we have added to the discussion as “Although we report a secondary outcome of hospital admissions as a marker of disease severity, data was not available for in-hospital treatment.” We are happy to consider further based on this revision whether the Reviewer and Editor feel this data strengthens the paper. We think on balance it is worth including these data as secondary outcomes, as it provides consistent findings with the primary outcome of COVID-19 mortality and therefore has important implication for health care utilisation.	9, 13-14, Figure 2,
BMI data from GDPPR were only available on approximately 50% of the cohort and as the authors discuss, this is a potential source of bias. Could the authors give a breakdown of the proportion in BMI categories by ethnicity and how this compares to other data on BMI in UK populations.	Thank you for this comment, which also aligns to the request by Reviewer 1 for additional stratification of the population characteristics by BMI status and ethnicity. The uploaded excel files therefore show the numbers by obesity status and ethnicity. In total 66.9% of people of White ethnicity 77.4% of Black ethnicity, 65.4% of South Asian ethnicity and 59.4% of other ethnicities were overweight or obese. Survey data from England in adults (18+ years) from 2015 to 2020 reported the same pattern of results, with rates of overweight or obesity highest in Black ethnic groups (between 67.5% to 73.6%), with rates in White ethnic group ranging from 62.0 to 63.7, with lower rates in Asian ethnic groups (ranging from 56.3 to 59.7%). https://www.ethnicity-facts-figures.service.gov.uk/health/diet-and-	18

	exercise/overweight-adults/latest The higher overall rates in our study compared to this survey are likely to reflect the older age group in our study (40+ years). We have extended the Discussion of the possible limitation of the missing data to include this point, as: “Nevertheless, the pattern of overweight and obesity in this study (66.9% for white ethnicities, 77.4% for black ethnicities, 65.4% for South Asian ethnicities and 59.4% for other ethnicities) were similar to national survey data where the highest rates (consistently above 70%) are reported in black ethnicities [31].”	
There is an impressive and extensive list of covariates included in the models (given in supp data). Were all the models stable with so many covariates and limited deaths in some groups? It would be helpful for some further discussion as to how these were selected for inclusion in the model (many would be correlated). Did the authors consider a model with a limited number of these covariates, including - for example - extant cardiometabolic or respiratory disease that are thought to be RF for COVID-19 severity? A number of the co-morbidities seem less relevant RF. A specific issue in this regard is whether the BMI association is independent of T2D, which is obviously common. Could the authors explore this further?	The list of covariates was selected specifically to align to the list of significant risk factors reported in the development of the QCOVID risk score (BMJ 2020; reference 20 in the text) which is consistent with the approach taken in ONS publications to date. Models were stable and it is worth noting the model fit for Model 1 (sociodemographic Census variables, including area and household deprivation, population density and key worker status) was already very high with a c-index of 0.902. The addition of clinical risk factors did not substantially improve model fit (c index = 0.920) or change the interpretation of the results (association between BMI and mortality). Therefore, the vast majority of potential confounding for this study is accounted for by the extensive list of sociodemographic census variables included within our model, which is a novel finding in itself. For diabetes, we included a combined diabetes (type 1 or type 2) within the model as one of the investigated chronic diseases. For the reviewers information, as the vast majority of cases were type 2 diabetes, results do not change (model fit stays the same) if the covariate selection is restricted to type 2 diabetes. As the reviewer noted, rates of diabetes as expected were highest in those who were obese (see included excel sheet of variables by ethnicity and BMI status). Finally, it is worth noting that the VIF was <3.5 at all modelled knots within the fully adjusted spline model, suggesting only modest collinearity and that only a modest proportion of the variance in BMI was explained by covariates including diabetes.	10
No definitions are given for the co-morbidities. Are these GP diagnoses and/or hospital diagnoses? Were they verified?	That you for highlighting this omission, we have added this detail to Table 1.	Table 1

Reviewer 3

Reviewer 3 Comment	Authors' Response	Page
A recent paper by Gao et al (Lancet Diabetes and Endocrinology, 2021) uses similar UK electronic health records data to analyse in some detail the links between BMI and COVID outcomes including mortality. The authors of that paper include an analysis of effect modification/interaction by ethnicity, finding evidence of interaction with a steeper BMI-COVID-death association in some non-white groups compared with white. The present paper is therefore not wholly novel. A limitation of the Gao paper is that they only looked at the interaction with the BMI-COVID-19 association specified as linear, whereas the current paper uses non-linear parameterisations and includes a useful visual of the interaction, in Figure 1. Another strength of the present paper is that linked census data were used to provide self-reported ethnicity, which is likely to be more accurate/complete than GP-recorded ethnicity.	Thank you for comment and discussion on our analysis vs the Gao paper. It also worth adding that the Gao paper only included patients during the first two months of the first wave of the pandemic in England (up to the end of April 2020), covering just 5479 deaths, with a low number of deaths reported in ethnic minority groups. In contrast, our analysis covers 33951 deaths, with over 1000 deaths in South Asian and Black ethnic groups, allowing for greater power, model stability, and precision. The Gao paper reported interactions by ethnicity as one of eight linear interaction models (i.e., explorative analysis), whereas ethnicity was the main focus of our paper. As the Reviewer highlights in their comment below, quantifying the risk between ethnic groups and how this varies across a continuous measure of BMI is importantly different to a focus simply on associations between BMI and COVID-19 mortality. We have added the following paragraph in the Introduction, which we feel makes this strength clearer. “However, whilst ethnicity has been shown to modify associations between BMI and COVID-19 outcomes, previous research has not quantified how this interaction affects both within-ethnicity and between-ethnicity risk across the spectrum of BMI. An early analysis of 5,623 community and in-hospital test results suggested the potential importance of this by showing the risk of SARS-CoV-2 positivity was not different between ethnic groups at low BMI, but was over two fold higher in minority ethnic groups compared to White ethnic group at high BMI [11]. This has not been explored in larger representative community cohorts or with COVID-19 outcomes.” We hope our response to the below comment and the resulting additional data analysis, graphs and Tables act to further highlights the novelty of our paper compared to previous analyses.	6
I found some of the analysis/presentation of results to be confusing/misleading. The stated aim was to look at how the BMI-	Thank you for this comment. We have clarified the research aim. It now reads: “The aim of this study was to use linked national Census, electronic health care records and mortality datasets to investigate the interaction between BMI and ethnicity in the risk of COVID-19 mortality, quantify how the	4, 7, 13-14, 16

COVID-death association varies by ethnicity, but in parts of the results the authors focus on the comparison between ethnic groups at fixed BMI levels, which addresses a different question. In addition, because all hazards are compared against the fixed reference point of "white - 22.5kg/m²", the BMI component is difficult to disentangle from the overall ethnic differences in risk. For example, the authors state that the HR for Black-30kg/m² compared with White-22.5 is 1.72 - but this will include both the effects of BMI and ethnicity, so does not really distill for us how the BMI-COVID-death association is different for Black ethnicity. The whole idea of "BMI of equivalent risk" suffers from the same flaw and I think is misleading. For example, it is stated that the equivalent risk to White-35kg/m² is seen at 25.2kg/m² in Black people. But this is not purely because of a steeper BMI-mortality slope in Black people (as implied by the conclusion that reducing levels of obesity is the solution to achieving equivalent risk) - the entire curve is also shifted substantially upwards due to the ethnicity	difference in risk between ethnic groups varies by BMI, and generate equivalency in risk at established BMI thresholds for class I, II, and III obesity” In our paper, we simultaneously presented the within and between ethnic risk across a continuous measure of BMI. However, we also agree that it is important to further consider and quantify how the relative risk of COVID-19 mortality varies by BMI (e.g. how much of the upward shift in risk in minority ethnic groups is influenced by BMI). We have reformatted Figure 1 and Figure 2 to display this further. Specifically, we have added an extra panel showing how the risk of COVID-19 mortality (and hospitalisation) in ethnic minority group compared to White Europeans varies across the range of continuous BMI. We have also created an additional table, showing modelled risk at specific BMI values to aid interpretation of the graphs and reduce the amount of data in the text to aid readability. The Reviewer will see that at low BMIs, Black and other ethnic groups do not have an elevated risk of COVID-19 mortality relative to the White ethnic group. However, the difference in risk widens as BMI increases. Similarly, there is only a marginally higher risk of COVID-19 mortality in South Asians relative to Whites at low BMI, but with risk over 3 times greater at a BMI of 40 kg/m². We hope the additional graph and Table makes it easier to visualise that the greater risk in ethnic minority groups at higher BMI is predominately determined by a steeper dose-response curve than by an upward shift of the whole curve. It is also worth highlighting the fact that previous analyses reporting equivalent BMI thresholds for type 2 diabetes (reference 12-14 in the manuscript), which is also the approach that has been used to inform national (e.g. NICE) and international (e.g. WHO) guideline recommendations for ethnic specific thresholds have also demonstrated very similar patterns of results, particularly for South Asians where the curve is shifted up, but with less difference in risk at lower BMIs. We believe the important findings for COVID-19 mortality is the steepness of the curve in ethnic minority populations results in equivalent BMI values that are more widely separated than those for cardiometabolic disease. Finally, we would like to thank the Reviewer again for this comment. We believe the additional analysis have greatly strengthened our paper, allowing for greater novelty and interpretation of the results. We have also revised our Abstract, Results and Discussion to incorporate/emphasise these results further.	
---	--	--

effect, independently of BMI.														
Only 52% of individuals had BMI present - a useful descriptive comparison of those with/without BMI data is provided, confirming missingness to be not completely at random. Missing data is discussed by the authors as a limitation, but it would be good to see more specific discussion of the assumptions being made by the authors' approach of excluding those with missing data (in particular, that missingness is conditionally independent of outcome).	We thank the Reviewer for this comment and for the opportunity to explore this important topic further. In order address this comment we ran additional logistical models to investigate whether covariates and outcome predicted missingness. The pseudo R² and AUROC data from the models are displayed below.    Predictor Variable (outcome missing yes/no) Pseudo R² Area under the receiver operating characteristic (AUROC)     Region 0.2086 0.7827   + Ethnicity and all included covariates 0.2133 0.7928   +outcome (COVID-19 mortality) 0.2133 0.7928    This data shows that to a moderate degree missingness is predicted by region (data shown in Supplementary eFigure 2). Other combinations of covariates and exposures add very little additional predictive discrimination. Of note, the addition of outcome to the model had no impact, confirming that the outcome does not predict missingness when conditioned on covariates. As has been noted previously [reference 22 in the text], this is an important step in producing unbiased risk estimates. Given we agree this is an important observation, we have added this data to the manuscript as additional sensitivity and supplementary material	Predictor Variable (outcome missing yes/no)	Pseudo R ²	Area under the receiver operating characteristic (AUROC)	Region	0.2086	0.7827	+ Ethnicity and all included covariates	0.2133	0.7928	+outcome (COVID-19 mortality)	0.2133	0.7928	11-12
Predictor Variable (outcome missing yes/no)	Pseudo R ²	Area under the receiver operating characteristic (AUROC)												
Region	0.2086	0.7827												
+ Ethnicity and all included covariates	0.2133	0.7928												
+outcome (COVID-19 mortality)	0.2133	0.7928												
Exclusion of 5% of BMI records (i.e. <2.5th and >97.5th %ile) to avoid outliers seemed extreme to me. What values did these %iles correspond to? Some sensitivity analysis around this would be reassuring.	Given the use of routine primary care records, we took a data driven approach to removing implausible values (such as 0 values) which are inevitable in such datasets. The 2.5th percentile corresponded to a value of 17.4 kg/m² and 97.5th percentile corresponded to a value of 41.0 kg/m². We have added these values to the text. For the reviewers information, the pattern of association was unchanged if all BMI data was included in the spline models, see below. We have not added this data to the revision, but are happy to consider further based on the review of this response.	9												

Death with COVID or suspected COVID anywhere on the death certificate was counted as a COVID-19 death. Was any sensitivity analysis done to explore other definitions (e.g. only confirmed COVID, only as underlying cause of death)?	The vast majority of deaths (31899/33951; 94.0%) were coded as U07.1 (lab-confirmed COVID-19 is mentioned as an underlying cause of death). Therefore, the removal of U07.2 coded deaths from the dataset did not change the pattern of results. However, a priori we thought it would be important to capture U07.2 deaths in the analyses due to the risk that confirmed COVID-19 was less likely at the start of the pandemic when testing was more limited (Pillar 1 in England). We have included the number of U07.1 deaths in the revised Results section.	13
Figure 2 appears to be essentially a repeat of Figure 1 with some additional annotation; it is not clear that this warrants an extra figure.	The Reviewer is correct – the Figure repeats Model 1 but adds the equivalence lines to aid interpretation. We included this Figure in the main text because we feel it provides a valuable additional tool to visualise the creation of the equivalent values which are likely to form an important part of the wider impact of this paper. However, we are happy to consider further on review of these revisions and to place in supplement if deemed appropriate by the Reviewer and the Editorial team.	

REVIEWERS' COMMENTS

Reviewer #1 (Remarks to the Author):

This is an important study on ethnicity, BMI and COVID-19 deaths, in a large population-based sample in the UK that uses electronic health records.

The design and analyses are elegant. The results are of broad generalizability globally. The authors addressed well the reviewer's comments and the revised manuscript has improved substantially.

I have few minor comments:

In the abstract please define that you present standard deviation next to \pm as the abstract should be a stand-alone part, regardless of the methods/other sections of the manuscript.

Abstract: Please add an explanation at least at first occurrence that the values in parenthesis represent 95% CI, for example 1.74 (95% CI 1.35, 2.26)

Covariates: table 1 does not include the variables age and sex. Please add an operational definition of the variable age: a main confounder and predictor of COVID-19 mortality. How it was analyzed in the multivariable models: as a categorical variable (which categories were used) or as a continuous variable? Please clarify this point in the manuscript.

The Results section lines 260-265: the description of figure 1: the association has a J shape, and not a "dose response" (similarly in the discussion lines 308-309). Please modify. Similarly in lines 278-279, figure 2 shows J shape associations especially in whites

HR: please define the abbreviation at first occurrence (both in the abstract and text of the manuscript).

Table 2 includes a lot of information in 4 panels, more details/explanations are needed (e.g., in footnotes) to improve the understanding of this complex table.

Figure 3: the dotted lines added to the figure together with BMI values does not improve the presentation of these data, rather it make it more complex. It is better adding these results just in text or a table.

Throughout the manuscript I believe that using "data are" rather than "data is", is preferable, but maybe this the Journal's style.

Figures 1 & 2: please indicate in the legend that the shaded areas represent 95% CI (as shown in the supplementary figures).

Supplementary material:

e-Tables 2 and 3: please add needed explanations/legend to this table. Please add units of measurement to the variable "Population density".

e-figure 4 presents the associations in women, but in the footnotes the authors mention that the models adjusted for sex. Similarly e-figure 5. Please revise.

Reviewer #2 (Remarks to the Author):

Thank you the opportunity to review this revised manuscript. Apologies for the slight delay in getting comments back.

The authors have done a thorough and thoughtful job of addressing my queries and those of the other reviewers, whose comments were insightful. I am not sure the authors fully addressed the issue of whether the T2D associations increased risk independently of BMI, but perhaps the editors can adjudicate.

I have no further comments.

Reviewer #3 (Remarks to the Author):

This is a thorough revision and I think the reviewers have taken on board and addressed all the main points satisfactorily. A couple of minor comments only:

Results p15 (track changes document): I suggest removing the sentence "The steeper dose-response association between BMI and COVID-19 mortality and the elevated risk in ethnic minority populations generated marked differences in equivalent BMI thresholds." Firstly this is moving into interpretation/discussion within the results section. But also, it seems to imply that the differences in slope cause different thresholds, yet we would have different BMI thresholds even if the lines were parallel (albeit that these may be exaggerated by the different slopes).

Discussion p17 (track changes document): A similar point regarding the following passage in the opening paragraph of the discussion: "The elevated risk of COVID-19 mortality in ethnic minority groups and steeper dose-repose with BMI resulted in an equivalent level of risk occurring at substantially lower BMI values". Again, the equivalent level of risk would have occurred at lower BMI values in ethnic minority groups even if there was no interaction and the lines were perfectly parallel. I suggest to remove this statement.

eTable 2: I believe that this table with descriptive data on the study population, by ethnicity, has been demoted to the supplementary. If possible, I think it would be preferable to have this in the main paper so that the reader can easily get a feel for the population under study. (see also comment below about redundant figures).

Figures 2/3: This is ultimately an editorial decision, but my view remains that only one of figures 2 and 3 is needed. If the authors wish to show the "BMI equivalent" values then Figure 3 alone would suffice, as this also includes all the information from Figure 2.

We thank the Editor and Reviewers for their positive feedback on our response and for the additional comments, which we have addressed below

REVIEWER COMMENT	AUTHOR RESPONSE
Reviewer 1	
In the abstract please define that you present standard deviation next to \pm as the abstract should be a stand-alone part, regardless of the methods/other sections of the manuscript. Abstract: Please add an explanation at least at first occurrence that the values in parenthesis represent 95% CI, for example 1.74 (95% CI 1.35, 2.26)	We have had to revise our Abstract to 150 words to fit with the journal style, so much detail has been lost and only the key non-technical elements retained
Covariates: table 1 does not include the variables age and sex. Please add an operational definition of the variable age: a main confounder and predictor of COVID-19 mortality. How it was analyzed in the multivariable models: as a categorical variable (which categories were used) or as a continuous variable? Please clarify this point in the manuscript.	Thank you. We have added these details to Table 1 and clarified age was analysed as a continuous variable.
The Results section lines 260-265: the description of figure 1: the association has a J shape, and not a "dose response" (similarly in the discussion lines 308-309). Please modify. Similarly in lines 278-279, figure 2 shows J shape associations especially in whites	We have removed reference to "dose-response" and replaced with "J-shaped" within the results, and removed reference to "steeper dose-response" in the Discussion and replaced simply with "stronger association"
HR: please define the abbreviation at first occurrence (both in the abstract and text of the manuscript).	This has been added to page 7
Table 2 includes a lot of information in 4 panels, more details/explanations are needed (e.g., in footnotes) to improve the understanding of this complex table.	We have added additional footnotes to Table 2 to aid interpretation
Figure 3: the dotted lines added to the figure together with BMI values does not improve the presentation of these data, rather it make it more complex. It is better adding these results just in text or a table.	Thanks you for this comment, which reinforces comments from reviewer 3. We have therefore added this Figure to supplement so it does not distract the reader in the main text.
Throughout the manuscript I believe that using "data are" rather than "data is", is preferable, but maybe this the Journal's style.	We agree and have revised through out
Figures 1 & 2: please indicate in the legend that the shaded areas represent 95% CI (as shown in the supplementary figures).	This has been added as requested
Supplementary material: e-Tables 2 and 3: please add needed	This has been clarified as people per square kilometre

explanations/legend to this table. Please add units of measurement to the variable "Population density".	
e-figure 4 presents the associations in women, but in the footnotes the authors mention that the modes adjusted for sex. Similarly e-figure 5. Please revise.	Thank you for spotting this error, now corrected
Reviewer 2	
The authors have done a thorough and thoughtful job of addressing my queries and those of the other reviewers, whose comments were insightful. I am not sure the authors fully addressed the issue of whether the T2D associations increased risk independently of BMI, but perhaps the editors can adjudicate.	Thank you for this remaining query. In our first response we focused on whether the reported associations are independent of T2D. For example our results are not affected after adjustment for diabetes. However, our analysis plan and stated objectives did not allow for an analysis as to whether diabetes was associated with higher risk independently of BMI. Whilst we agree this would be an interesting and worthwhile question in its own right, it is perhaps outside the scope of the current paper.
Reviewer 3	
Results p15 (track changes document): I suggest removing the sentence "The steeper dose-response association between BMI and COVID-19 mortality and the elevated risk in ethnic minority populations generated marked differences in equivalent BMI thresholds." Firstly this is moving into interpretation/discussion within the results section. But also, it seems to imply that the differences in slope cause different thresholds, yet we would have different BMI thresholds even if the lines were parallel (albeit that these may be exaggerated by the different slopes).	Thank you for this additional feedback. We removed the sentence as suggested. We have also reordered slightly so this now follows on from the other results for COVID-19 mortality as without the opening sentence the paragraph does not stand alone as well.
Discussion p17 (track changes document): A similar point regarding the following passage in the opening paragraph of the discussion: "The elevated risk of COVID-19 mortality in ethnic minority groups and steeper dose-repose with BMI resulted in an equivalent level of risk occurring at substantially lower BMI values". Again, the equivalent level of risk would have occurred at lower BMI values in ethnic minority groups even if there was no interaction and the lines were perfectly parallel. I suggest to remove this statement.	In the Discussion we feel it is important to retain some interpretation of the findings. Whilst we agree absolutely with the Reviewer comment, we were referring specifically to the pattern of associations observed in this study, which were partly driven by the interaction. In other words the equivalent values become more separated the higher the BMI. However, in order to avoid confusion, we have revised to (p10): "The pattern of association between BMI and COVID-19 mortality across ethnic groups produced in an equivalent level of risk at substantially different BMI values".

eTable 2: I believe that this table with descriptive data on the study population, by ethnicity, has been demoted to the supplementary. If possible, I think it would be preferable to have this in the main paper so that the reader can easily get a feel for the population under study. (see also comment below about redundant figures).	The Reviewer is correct this Table was moved to Supplement on the advice of Reviewer 1 during the first revision. We agree with reviewer 1 that due to its size, it will possibly be distracting to the reader to have this embedded within the main text. However, we are happy for the Editor to adjudicate and move as required.
Figures 2/3: This is ultimately an editorial decision, but my view remains that only one of figures 2 and 3 is needed. If the authors wish to show the "BMI equivalent" values then Figure 3 alone would suffice, as this also includes all the information from Figure 2.	It would appear the Reviewers (see Reviewer 1) have consensus that this combination of Figures is not optimal. We have therefore retained Figure 2, but moved Figure 3 to Supplement